# SARS-CoV-2-Specific Antibodies, B Cell and T Cell Immune Responses after ChAdOx1 nCoV-19 Vaccination in Solid Organ Transplant Recipients

**DOI:** 10.3390/vaccines12050541

**Published:** 2024-05-15

**Authors:** Pattaraphorn Phornkittikorn, Surasak Kantachuvesiri, Abhasnee Sobhonslidsuk, Teerapat Yingchoncharoen, Sasisopin Kiertiburanakul, Jackrapong Bruminhent

**Affiliations:** 1Department of Medicine, Faculty of Medicine Ramathibodi Hospital, Mahidol University, Bangkok 10400, Thailand; ll.zeitgeist.ll@windowslive.com (P.P.); surasak.kan@mahidol.ac.th (S.K.); abhasnee.sob@mahidol.ac.th (A.S.); teerapatmdcu@gmail.com (T.Y.); sasisopin@hotmail.com (S.K.); 2Ramathibodi Excellence Center for Organ Transplantation, Faculty of Medicine Ramathibodi Hospital, Mahidol University, Bangkok 10400, Thailand

**Keywords:** SARS-CoV-2, COVID-19, COVID-19 vaccines, immunocompromised, organ transplant, immunity, spike protein, receptor-binding domain

## Abstract

Background: Immunization against SARS-CoV-2 is essential for vulnerable solid organ transplant (SOT) recipients who are at risk of infection. However, there are concerns about suboptimal immunogenicity, especially in humoral immunity (HMI), and limited exploration of cell-mediated immune (CMI) responses. The primary objective of this study was to assess the immunogenicity of ChAdOx1 nCoV-19 vaccination in SOT recipients. The secondary endpoint was to evaluate factors that affect immunogenicity and adverse events (AEs) following immunization in SOT recipients. Methods: All adult SOT recipients who received the two-dose ChAdOx1 nCoV-19 vaccine at a 12-week interval underwent measurements of HMI by evaluating anti-receptor-binding domain (RBD) IgG levels and CMI by investigating SARS-CoV-2-specific T cell and B cell responses before and after complete vaccination, around 2–4 weeks post-vaccination, and compared to controls. AEs were monitored in all participants. Results: The study included 63 SOT recipients: 44 kidney recipients, 16 liver recipients, and 3 heart transplant recipients, along with 11 immunocompetent controls. Among SOT recipients, 36% were female, and the median (IQR) age was 52 (42–61). The median (IQR) time since transplant was 55 (28–123) months. After the second dose, the median (IQR) anti-RBD antibody levels were significantly lower in SOT recipients compared to those in the control group (8.3 [0.4–46.0] vs. 272.2 [178.1–551.6] BAU/mL, *p* < 0.01). This resulted in a seroconversion rate (anti-RBD antibody > 7.1 BAU/mL) of 51% among SOT recipients and 100% among controls (*p* = 0.008). Receiving the vaccine beyond one year post-transplant significantly affected seroconversion (OR 9.04, 95% CI 1.04–78.56, *p* = 0.046), and low-dose mycophenolic acid marginally affected seroconversion (OR 2.67, 95% CI 0.89–7.96, *p* = 0.079). RBD-specific B cell responses were also significantly lower compared to those in the control group (0 [0–4] vs. 10 [6–22] SFUs/10^6^ PBMCs, *p* = 0.001). Similarly, S1- and SNMO-specific T cell responses were significantly lower compared to those in the control group (48 [16–128] vs. 216 [132–356] SFUs/10^6^ PBMCs, *p* = 0.004 and 20 [4–48] vs. 92 [72–320] SFUs/10^6^ PBMCs, *p* = 0.004). AEs were generally mild and spontaneously resolved. Conclusions: SOT recipients who received the full two-dose ChAdOx1 nCoV-19 vaccine demonstrated significantly diminished HMI and CMI responses compared to immunocompetent individuals. Consideration should be given to administering additional vaccine doses or optimizing immunosuppressant regimens during vaccination (Thai Clinical Trial Registry: TCTR20210523002).

## 1. Introduction

In recent years, the world has faced the challenge of the COVID-19 pandemic, which is caused by coronavirus disease 2019 (COVID-19). This disease can present with a wide spectrum of symptoms, ranging from asymptomatic cases to severe respiratory failure. In an effort to mitigate the potential severity of the disease and reduce associated complications, vaccination against severe acute respiratory syndrome coronavirus 2 (SARS-CoV-2) is strongly recommended [1].

Solid organ transplant (SOT) recipients necessitate lifelong immunosuppressive therapy. The prolonged use of immunosuppressant medications is crucial for enhancing graft survival in transplant recipients. Meanwhile, combining multiple immunosuppressive drugs still causes a significant risk for various infections because of dramatically reduced immunity against viruses, bacteria, and other pathogens [2,3,4,5]. Therefore, SOT recipients are vulnerable to severe COVID-19 due to an immunosuppressive state [6].

Immunization against SARS-CoV-2 is essential and proven to avoid complications and unfavorable consequences [6,7]. However, suboptimal immunogenicity, especially humoral immunity (HMI), is concerning among these specific individuals [8,9]. Although cell-mediated immunity was believed to play an essential role in preventing the progression of the disease, it has been reported to vary among different vaccine platforms [10,11,12,13,14]. These could affect the memory immune cells, which rely on the T cell function to elicit. Furthermore, cell-mediated immune (CMI) responses, especially B cell-specific immunity, have not been explored much. We believe that the cell-mediated immune response, especially memory B cells, could play an essential role in immunogenicity after vaccination, particularly in providing long-term protection against infection.

As such, our study aimed to evaluate the immunogenicity of the two-dose ChAdOx1 nCoV-19 vaccine regimen in SOT recipients, focusing on their HMI, CMI, and, notably, B cell-specific immunity. Our primary objective was to assess the immunogenicity following ChAdOx1 nCoV-19 vaccination in SOT recipients. In addition, our secondary objective was to investigate the factors influencing immunogenicity in SOT recipients and to monitor any adverse events (AEs) that occurred following immunization.

## 2. Material and Methods

### 2.1. Study Design

A prospective cohort study was conducted from November 2021 to January 2022, involving solid organ transplant (SOT) recipients, including kidney, liver, and heart transplant recipients, at the Excellence Center for Organ Transplantation within the Faculty of Medicine at Ramathibodi Hospital, Mahidol University, Bangkok, Thailand.

In our study, we conducted an assessment of SARS-CoV-2-specific immune responses both before the administration of the first vaccine dose and 2–4 weeks after the second dose. Specifically, we evaluated SARS-CoV-2-specific humoral immunity (HMI) using a SARS-CoV-2 immunoglobulin G (IgG) assay, which detects antibodies directed against the receptor-binding domain (RBD) of the SARS-CoV-2 spike protein. Additionally, we assessed SARS-CoV-2-specific cell-mediated immunity (CMI) responses by measuring interferon-γ (IFN-γ)-producing T and B cell responses through an enzyme-linked immunospot (ELISpot) assay.

Furthermore, we conducted an analysis to identify predictors of anti-RBD antibody seroconversion. In terms of safety, we closely monitored AEs occurring within 3 and 7 days following vaccination. This monitoring was carried out through direct contact, involving phone calls and reports provided by the participants afterward.

### 2.2. SARS-CoV-2-Specific Humoral Immune Responses

HMI was evaluated using the Abbott SARS-CoV-2 IgG II Quantification assay (Abbott, Sligo, Ireland), which is a chemiluminescent microparticle immunoassay designed for the quantitative detection of IgG antibodies in human serum. This particular assay is specifically geared towards measuring IgG antibodies that are specific to the receptor-binding domain (RBD) of the SARS-CoV-2 spike protein. The ARCHITECT i2000SR system (Singapore) was used to conduct the Abbott assay. The results for anti-RBD IgG were quantified in binding arbitrary units (BAU) per mL. The company provided conversion factors to compute WHO BAU/mL: 1 BAU/mL, equating to 0.142 AU/mL. A quantitative outcome reaching or surpassing 7.1 BAU/mL was regarded as indicative of seroconversion [15].

### 2.3. SARS-CoV-2-Specific Cell-Mediated Immune Responses

The evaluation of CMI involved the measurement of IFN-γ-producing T and B cell responses through an ELISpot assay. In this assay, IFN-γ production was assessed using a human IFN-γ ELISpot PRO kit and activated peripheral blood mononuclear cells (PBMCs). PBMCs were stimulated under various conditions, including a negative control, the SARS-CoV-2 S1 domain of the spike protein scanning peptide pool, the SNMO peptide pool (consisting of SARS-CoV-2 spike protein, nucleoprotein, membrane protein, open reading frame [ORF]-3a, and ORF-7a proteins), and anti-CD3 antibodies as a positive control. Following incubation, the cells were removed, and the production of IFN-γ was determined. The results were reported as the median and interquartile range (IQR) of IFN-γ-producing spot-forming units (SFUs) per 10^6^ PBMCs for each peptide pool [16]. Additionally, anti-SARS-CoV-2 RBD IgG antibody-secreting cells (memory B cells) were measured using ELISpot assays with a Human IgG (SARS-CoV-2, RBD) ALP (Mabtech). The results were reported as B cells secreting anti-SARS-CoV-2 RBD IgG and those secreting any IgG (total IgG) [16]. The emerged spots were counted using an ImmunoSpot analyzer from Cellular Technology Limited, located in Shaker Heights, OH, USA, and spot quality was assessed using ImmunoSpot Software v5.0.9.15.

### 2.4. Safety

Prior to vaccination, the patients underwent a thorough check of their vital signs and a physical examination. Following vaccination, immediate adverse events (AEs) were closely monitored for up to 30 min. Additionally, phone calls were conducted at the 3-day and 7-day marks after each vaccination to monitor any solicited AEs. Patients were encouraged to report any unsolicited AEs as well. Furthermore, participants were advised to contact the healthcare facility if they developed any respiratory symptoms, required advice, or needed medical attention for further evaluation of AEs or for a potential COVID-19 diagnosis. If any participant received a confirmed COVID-19 diagnosis via nasopharyngeal or oropharyngeal swab, they received treatment in accordance with the standard of care. Participants were instructed to reach out to the investigators with any concerns after 7 days of vaccination.

### 2.5. Data Collection

Our study enrolled SOT recipients who met the following criteria: they were 18 years of age or older, had undergone transplantation at least one month prior, and had maintained stable allograft function and immunosuppressive regimens.

Conversely, individuals who fell into the following categories were excluded from our study: those who had symptoms suggestive of a respiratory tract infection within the preceding 3 days, those currently experiencing an active infection, individuals recently diagnosed with allograft dysfunction necessitating alterations to their immunosuppressive regimens, those with a prior history of COVID-19 infection, participants who had received another vaccination within the past 4 weeks, or those who had previously received a different COVID-19 vaccine.

Baseline and transplant characteristics were collected comprising age, gender, organ transplant type, allograft type, onset after transplant, immunosuppressive regimen, and dosing. The classification of a low C_0_ level of calcineurin inhibitors (CNIs) was either cyclosporine of 150 or less ng/mL or tacrolimus of 5 or less ng/mL [17]. The classification of a low therapeutic dose of mycophenolic acid (MPA) was either mycophenolate sodium (MPS) of 720 or less mg/day or mycophenolate mofetil (MMF) of 1 g or less/day [18].

### 2.6. Statistical Analysis

Continuous variables were summarized using medians and interquartile ranges, while categorical variables were summarized with frequencies and percentages. Participant characteristics and immunogenicity following two doses of the ChAdOx1 nCoV-19 vaccine were described using descriptive statistics. The assessment of differences between groups was performed using Fisher’s exact test and the Mann–Whitney U test. Univariate analysis using logistic regression analysis was employed to identify predictors affecting the seroconversion rate in solid organ transplant (SOT) recipients. Statistical significance was defined as a *p*-value less than 0.05. All statistical analyses were conducted using the Statistical Package for Social Sciences software program (SPSS^®^ Statistics 18 (IBM, Armonk, NY, USA)). The study protocol received approval from the Human Research Ethics Committee of the Faculty of Medicine at Ramathibodi Hospital, Mahidol University, in Bangkok, Thailand. Additionally, all participants provided their informed consent before enrolling in the study (Ethics Committee Reference: MURA2021/401). The study was registered with the Thai Clinical Trial Registry under the identifier TCTR20210523002.

## 3. Results

### 3.1. Study Population

After excluding 17 SOT recipients due to a history of COVID-19 infection during the study period and a change in participation decision because of inconvenient transportation during the epidemic, we excluded one further immunocompetent participant due to missing data. A total of 63 SOT recipients, including 44 kidney, 16 liver, and 3 heart transplant recipients and 9 immunocompetent controls, were recruited into this study as shown in the study flow in Figure 1.

Patient characteristics are presented in Table 1. Among the SOT recipients, 36% were female, and the median (IQR) age was 52 (42–61). The median (IQR) time since transplant was 55 (28–123) months. Controls included 11 immunocompetent people: 2% were male, and the median (IQR) age was 37 (29–45). Compared to the participants, the control groups were younger and had a lesser proportion of male participants (52 [42–61] vs. 37 [29–45] years old, *p* = 0.01 and 40 vs. 2, *p* = 0.03, respectively).

### 3.2. Immunogenicity

After the second dose, the median (IQR) of the anti-RBD antibody levels was significantly lower in SOT recipients compared to those of the controls (8.3 [0.4–46.0] vs. 272.2 [178.1–551.6] BAU/mL, *p* < 0.01), which resulted in a rate of seroconversion (anti-RBD antibody ≥ 7.1 BAU/mL) of 51% and 100%, respectively (*p* = 0.008) (Figure 2). For sensitivity analysis, by excluding heart transplant recipients, the median (IQR) of the anti-RBD antibody levels remained significantly lower in SOT recipients compared to those of the controls (2.2 [0.3–6.9] vs. 272.2 [178.1–551.6] BAU/mL, *p* < 0.01).

The predictors for seroconversion are presented in Figure 3. In the univariate analysis, receiving the vaccine beyond one year post-transplant was significantly associated with an increased likelihood of seroconversion (odds ratio [OR] 9.04, 95% confidence interval [95% CI] 1.04–78.56, *p* = 0.046). Notably, the use of low-dose mycophenolic acid exhibited a marginally significant impact on individuals who did not achieve seroconversion (OR 2.67, 95% CI 0.89–7.96, *p* = 0.079).

S1- and SNMO-specific T cell responses were also significantly lower compared to those of the controls (48 [16–128] vs. 216 [132–356] SFUs/10^6^ PMBCs, *p* = 0.004 and 20 [4–48] vs. 92 [72–320] SFUs/10^6^ PMBCs, *p* = 0.004), as seen in Figure 4 and Table 2. In parallel, RBD-specific B cell responses were also significantly lower compared to those of the controls (0 [0–4] vs. 10 [6–22] SFUs/10^6^ PMBCs, *p* = 0.001), as shown in Figure 5.

### 3.3. Safety

AEs were generally mild and spontaneously resolved (Table 3 and Table 4). Regarding solicited AEs of the first dose of vaccine, forty-nine (78%) and four (6%) patients experienced AEs on day 3 and day 7, respectively. Examples of AEs included pain at the injection site, fever, and muscle aches. Regarding unsolicited AEs of the first dose of vaccine, three (5%) and one (2%) patients experienced AEs on day 3 and day 7, respectively. Unsolicited AEs found in the patients included chest pain and syncope. No AEs were reported by the participants for up to six months.

## 4. Discussion

A prospective study aimed at evaluating immunogenicity in Thai solid organ transplant (SOT) recipients who received a two-dose viral-vectored vaccine with a 12-week interval revealed markedly diminished humoral immune (HMI) and cellular immune (CMI) responses in comparison to individuals with immunocompetent status. Notably, SOT recipients who were on high-dose mycophenolic acid maintenance therapy exhibited a lower rate of seroconversion. Nevertheless, it is worth highlighting that the vaccine demonstrated a satisfactory safety profile and tolerability, as evidenced by a relatively short-duration monitoring period.

SOT recipients are recognized as a particularly vulnerable group, facing a heightened risk of experiencing severe manifestations following a COVID-19 infection. In the context of the multiple COVID-19 vaccine platforms available in Thailand at that time, there remained a paucity of concrete evidence regarding the efficacy of these vaccines in SOT recipients. Moreover, there was a notable absence of data concerning the immunogenic response within this specific patient population.

Our results showed that about 51% of SOT recipients were seroconverted following the two-dose ChAdOx1 nCoV-19 vaccine, while 100% of healthy controls were seroconverted. The result of immunogenic response in SOT recipients after receiving a two-dose ChAdOx1 nCoV-19 vaccine revealed that the development of HMI, indicated by anti-RBD IgG levels, was not adequately achieved and was poor compared with healthy participants. CMI responses appeared in the same direction as HMI [19]. The findings of our study imply that SOT recipients may require more than two doses of the ChAdOx1 nCoV-19 vaccine to achieve effective prophylaxis. It is worth noting that low-dose mycophenolic acid exhibited a marginally significant impact, which could potentially become more pronounced with an increase in the study’s statistical power. Additionally, the timing of vaccination beyond one year post-transplant was significantly associated with seroconversion. The favorable predictors of seroconversion observed in our study align with the trends reported in other studies.

The novelty of our research lies in the exploration of the potential impact of memory B cells, among other cell-mediated immune responses, on post-vaccination immunogenicity. This aspect, crucial for long-term protection against infection, has received limited attention in previous studies.

The studies reported that high therapeutic doses of MPA can blunt the immune response in SOT recipients [8,19,20,21]. In a French cohort study, shorter post-transplant duration and higher levels of maintenance immunosuppressants were predictive of seroconversion failure in kidney transplant recipients [22,23]. In an Austrian cohort study investigating immunogenicity after an additional vaccine, it was found that KT recipients who had undergone the procedure over a year ago showed a notably increased probability of seroconversion [21]. These characteristics could be utilized in the development of a personalized COVID-19 immunization strategy based on individual immune status. Moreover, a temporary discontinuation of immunosuppressants during immunization holds promise for enhancing immune responses, provided that the risk of allograft rejection is manageable [24].

In a comprehensive analysis, it was found that the cellular immune response post-SARS-CoV-2 vaccination in KT recipients was inferior when compared to dialysis patients and immunocompetent individuals [25]. More evidence shows that kidney transplant recipients need an extra vaccine dose to boost their T and B cell immune responses [26].

The assessment of safety outcomes following the vaccine regimen offered to participants was a concern. AEs during the early period at 3-day and 7-day post-vaccination were reported to be mild, as demonstrated. The observed AE profile is like those reported in previous COVID-19 vaccine studies [27,28]. The most common AE reported in our study was injection site pain, reported by approximately half of SOT recipients who obtained vaccination. However, long-term AEs and allograft dysfunction rates still need further follow-up.

Although our transplant center successfully transplanted many SOT recipients each year, we still had fewer participants than expected. After starting this study for several months, some participants decided to switch the vaccine to another platform because of concerns about the efficacy of a particular COVID-19 vaccine. Meanwhile, some subsequently changed their decision to not participate due to complex measures for traveling across the province during the epidemic. This could potentially restrict the study’s power due to an unexpectedly smaller participant pool, especially among specific transplant types such as heart transplant recipients. Furthermore, the absence of SARS-CoV-2 antibody testing to exclude previous COVID-19 infection could potentially affect the seroconversion rate. Ideally, anti-nucleocapsid antibodies should have been measured prior to immunization in order to exclude a surge in immunity from natural infection. However, we promptly provided the vaccine to our participants during a time when COVID-19 had not yet been widely spread in the community. Hence, there was a low chance of natural infection occurring. In addition, our data might not be applicable to continuously evolving viral strains. Moreover, reports have indicated a decrease in antibody levels after 3–8 weeks following the second vaccination in patients who received ChAdOx1 nCoV-19 or BNT162b2 [10]. With regard to the control group, we acknowledge a disparity in gender distribution, significant age gaps, and a restricted number of participants. These factors hinder our ability to conduct an appropriate sex-matched case–control study design.

## 5. Conclusions

In summary, our study indicates that SOT recipients who received the complete two-dose regimen of the ChAdOx1 nCoV-19 vaccine exhibited significantly lower levels of HMI and CMI responses in comparison to immunocompetent individuals. To enhance the vaccine’s efficacy in this population, it may be advisable to explore the administration of additional vaccine doses or the optimization of immunosuppressant medications during the vaccination process. Furthermore, a booster dose to maintain adequate immunity may be warranted in these vulnerable populations. It is reassuring to note that short-term AEs were observed among SOT recipients but were generally well tolerated.

## Figures and Tables

**Figure 1 vaccines-12-00541-f001:**
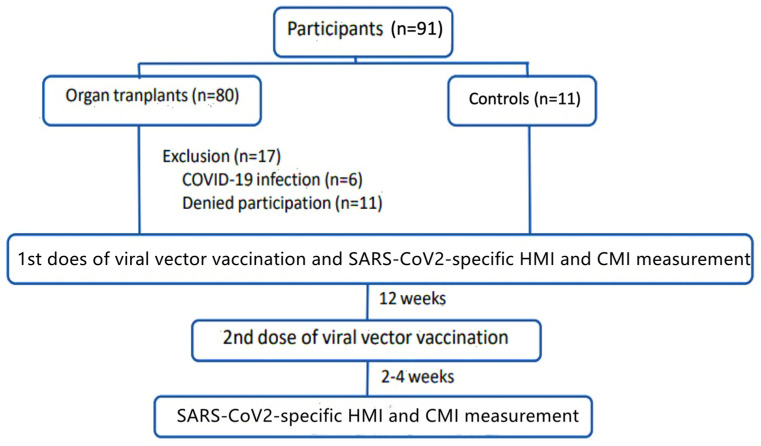
Study flow.

**Figure 2 vaccines-12-00541-f002:**
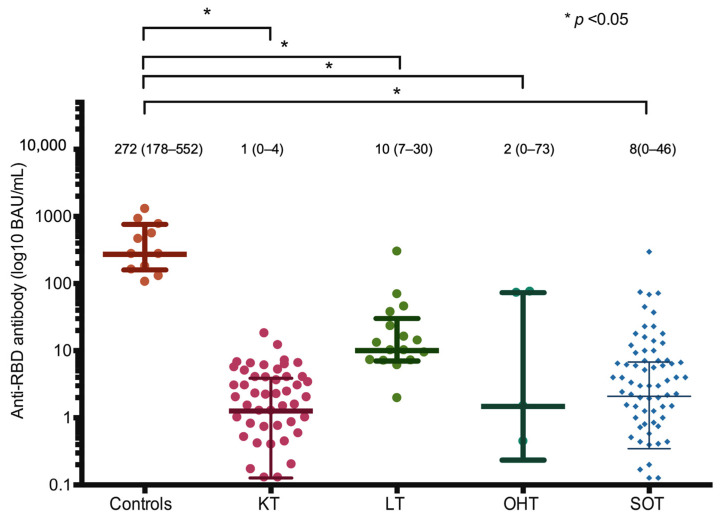
SARS-CoV-2 RBD-specific IgG antibody level 2–4 weeks post-second dose in healthy controls and SOT recipients. Bar represents IQR with the median (IQR) value. * *p* value < 0.05. Abbreviations: BAU, binding antibody unit; SOT, solid organ transplant; KT, kidney transplant; LT, liver transplant; OHT, orthotopic heart transplant; RBD, receptor-binding domain.

**Figure 3 vaccines-12-00541-f003:**
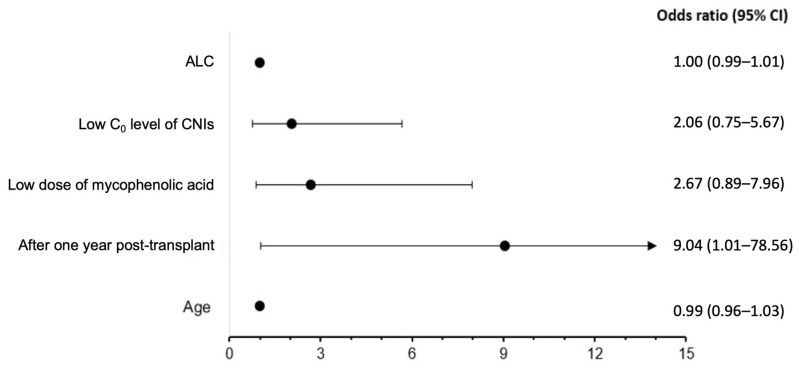
Predictors effect on seroconversion rate in SOT recipients.

**Figure 4 vaccines-12-00541-f004:**
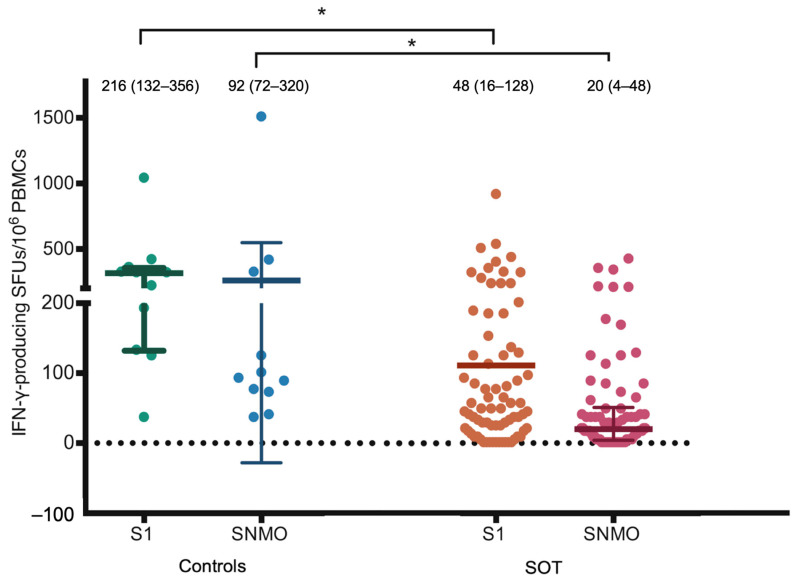
SARS-CoV-2-specific IFN-γ-producing T cell responses reactive to the S1 protein. SMNO protein detected by IFN-γ ELISpot assay 2–4 weeks post-second dose. Bar represents IQR with the median (IQR) value. * *p* value < 0.05. Abbreviations: IFN-γ, interferon-γ; SOT, solid organ transplant; SFUs, spot-forming units; PBMC, peripheral blood mononuclear cell; S, spike glycoprotein; S1, S1 domain of spike protein; S2N, spike and nucleoproteins; SNMO: peptide pool of spike protein, nucleoprotein, membrane protein, and open reading frame proteins.

**Figure 5 vaccines-12-00541-f005:**
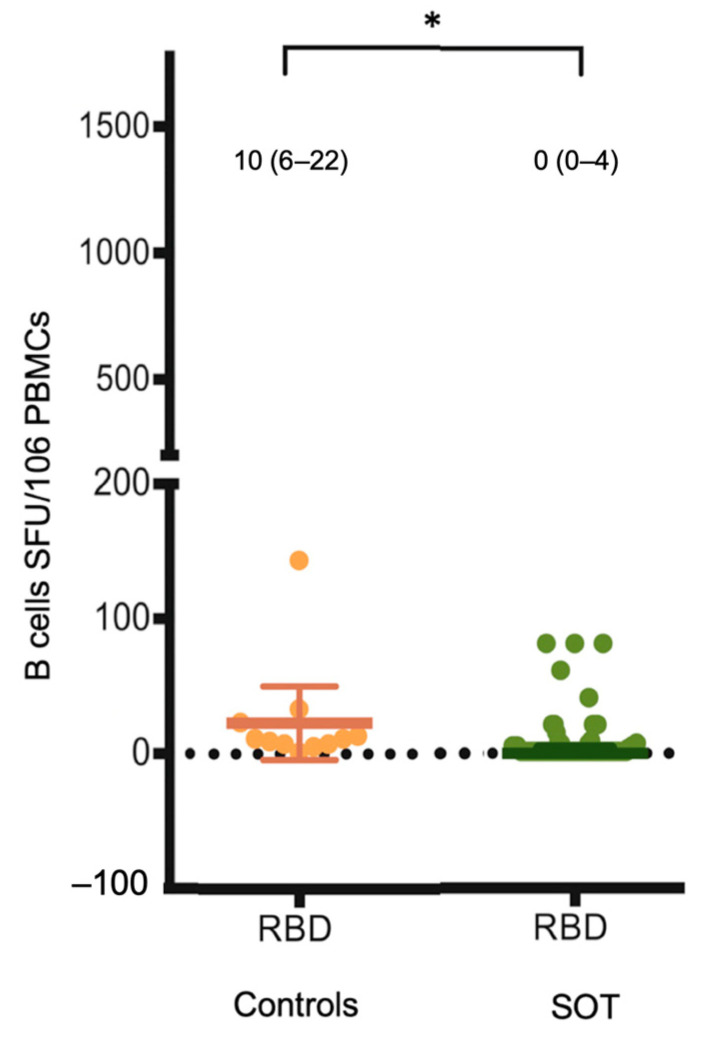
SARS-CoV-2-specific IFN-γ-producing B cell responses reactive to RBD detected by IFN-γ ELISpot assay 2–4 weeks post-second dose. Bar represents IQR with the median (IQR) value. * *p* value < 0.05. Abbreviations: IFN-γ, interferon-γ; SOT, solid organ transplant; SFUs, spot-forming units; PBMC, peripheral blood mononuclear cell; R, receptor-binding domain.

**Table 1 vaccines-12-00541-t001:** Patient characteristics.

Clinical Characteristic	SOT Recipients (n = 63)	Control (n = 11)	*p*-Value
Age, [years], median (IQR)	52 (42–61)	37 (29–45)	0.01
18–34 [years], n (%)	0 (0)	11 (100)	
35–59 [years], n (%)	23 (36)	0 (0)	
≥60 [years], n (%)	40 (64)	0 (0)	
Male sex, n (%)	40 (63)	2 (18)	0.03
Time from transplant to vaccination [months], median (IQR)	55 (28–123)		
Vaccination within one year post-transplant, n (%)	8 (12.7)		
Deceased allograft, n (%)	50 (79.4)		
Kidney transplant	44 (69.8)		
Heart transplant	3 (4.7)		
Liver transplant	16 (25.3)		
Immunosuppressant, n (%)			
Tacrolimus, n (%)	45 (71.4)		
C_0_ level [ng/mL], median (IQR)	5.2 (4.3–5.8)		
Cyclosporine, n (%)	17 (27)		
C_0_ level [ng/mL], median (IQR)	90 (51–118)		
Low C_0_ level of calcineurin inhibitors ^a^	26 (41.9)		
Mycophenolate mofetil, n (%)	41 (65)		
Dose [mg/day], median (IQR)	1000 (1000–1500)		
Mycophenolate sodium, n (%)	8 (12.7)		
Dose [mg/day], median (IQR)	900 (585–1080)		
Low therapeutic dose of mycophenolic acid ^b^	27 (43)		
Everolimus, n (%)	5 (7.9)		
Prednisolone, n (%)	45 (71.4)		
Dose [mg/day], median (IQR)	5 (5–5)		
Absolute lymphocyte count [cell/µL], median (IQR)	1800 (1300–2500)		

^a^ Dominator is the number of participants who received a calcineurin inhibitor (n = 62). ^b^ Dominator is the number of participants who received a mycophenolic acid (n = 49). Abbreviations: IQR, interquartile range; C_0_, initial plasma concentration of drug at time = 0; SOT, solid organ transplant.

**Table 2 vaccines-12-00541-t002:** Immunogenicity in SOT recipients after receiving two-dose ChAdOx1 nCoV-19 vaccine.

Immune Responses	SOT Recipients (n = 63)	Control (n = 11)	*p*-Value
Anti-RBD IgG [BAU/mL], median (IQR)	8.3 (0.4–46.0)	272.2 (178.1–551.6)	<0.01
Rate of seroconversion, n (%)	32 (51)	11 (100)	0.008
S_1_-specific T cells [SFUs/10^6^PBMCs], median (IQR)	48 (16–128)	216 (132–356)	0.004
SNMO-specific T cells [SFUs/10^6^PBMCs], median (IQR)	20 (4–48)	92 (72–320)	0.004
RBD-specific B cells [SFUs/10^6^PBMCs], median (IQR)	0 (0–4)	10 (6–22)	0.001

Abbreviations: BAU, binding antibody unit; IgG, immunoglobulin G; IQR, interquartile range; SOT, solid organ transplant; PBMC, peripheral blood mononuclear cell, RBD, receptor-binding domain; S1, S1 domain of spike protein; SARS-CoV-2, severe acute respiratory syndrome coronavirus 2; SFUs, spot-forming units.

**Table 3 vaccines-12-00541-t003:** Solicited adverse events after ChAdOx1 nCoV-19 vaccine in SOT recipients.

Solicited Adverse Events, n (%)	SOT Recipients (n = 63)	
	1st Dose	2nd Dose
Day 3		
Adverse events	49 (78)	32 (51)
Grade 1	49 (78)	32 (51)
Grade 2	0 (0)	0 (0)
Grade 3	0 (0)	0 (0)
Pain at injection site	34 (54)	25 (40)
Muscle aches	12 (19)	4 (6)
Increased appetite	3 (5)	1 (2)
Fever	15 (24)	7 (11)
Sleepiness	6 (10)	1 (2)
Others	38 (60)	17 (27)
Day 7		
Adverse events	4 (6)	2 (3)
Grade 1	4 (6)	2 (3)
Grade 2	0 (0)	0 (0)
Grade 3	0 (0)	0 (0)
Pain at injection site	1 (2)	0 (0)
Muscle aches	1 (2)	0 (0)
Increased appetite	1 (2)	0 (0)
Fever	0 (0)	0 (0)
Sleepiness	0 (0)	0 (0)
Others	4 (6)	2 (3)

**Table 4 vaccines-12-00541-t004:** Unsolicited adverse events after ChAdOx1 nCoV-19 vaccine in SOT recipients.

Unsolicited Adverse Events, n (%)	SOT Patients (n = 63)	
	1st Dose	2nd Dose
Day 3		
Adverse events	3 (5)	1 (2)
Grade 1	2 (3)	1 (2)
Grade 2	1 (2)	0 (0)
Grade 3	0 (0)	0 (0)
Day 7		
Adverse events	1 (2)	1 (2)
Grade 1	0 (0)	0 (0)
Grade 2	1 (2)	1 (2)
Grade 3	0 (0)	0 (0)

## Data Availability

The datasets generated during and/or analyzed during the current study are available from the corresponding author upon reasonable request.

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
