# Peer review of "SARS-CoV-2-Specific Antibodies, B Cell and T Cell Immune Responses after ChAdOx1 nCoV-19 Vaccination in Solid Organ Transplant Recipients"

_vaccines, 2024, doi:10.3390/vaccines12050541_

Round 1

Reviewer 1 Report (Previous Reviewer 2)

Comments and Suggestions for Authors

Referring to the previous version, this revised version seeks to enhance the overall quality and clarity. Additionally, this version put a suggestion for the administration of booster doses for the at-risk group.

Comments.

1. Figures 2, 4 and 5: Consider enhancing the informativeness and readability of Figures 2, 4, and 5 by adding the median and interquartile range (IQR) for each column. Displaying these values above the respective columns will provide readers with a clearer understanding of the data.

2. Age group: Why you used age group following 18-40, 40-60, and >60?

Almost widely recognised age categories, such as 18-34 (young adult), 35-59 (middle-aged), and ≥60 (elderly).
This adjustment aligns with common age classifications, promoting consistency and facilitating comparison with established literature.

Author Response

Referring to the previous version, this revised version seeks to enhance the overall quality and clarity. Additionally, this version put a suggestion for the administration of booster doses for the at-risk group.

Comments.

1. Figures 2, 4 and 5: Consider enhancing the informativeness and readability of Figures 2, 4, and 5 by adding the median and interquartile range (IQR) for each column. Displaying these values above the respective columns will provide readers with a clearer understanding of the data.Answer: thank you very much.

2. Age group: Why you used age group following 18-40, 40-60, and >60?

Almost widely recognised age categories, such as 18-34 (young adult), 35-59 (middle-aged), and ≥60 (elderly).
This adjustment aligns with common age classifications, promoting consistency and facilitating comparison with established literature.

Answer: thank you very much.

Reviewer 2 Report (Previous Reviewer 3)

Comments and Suggestions for Authors

Thank you for inviting me to review this manuscript. The authors chose to include all transplants in the same analysis. I am concerned that assuming all transplant recipients as a single group will significantly affect the results and conclusions. This concern could have been easily addressed by conducting a sensitivity analysis, which they neglected to do. Furthermore, they lacked the necessary power to perform a matched case-control study, and the logistic regression conducted was merely univariable.

Author Response

Thank you for inviting me to review this manuscript. The authors chose to include all transplants in the same analysis. I am concerned that assuming all transplant recipients as a single group will significantly affect the results and conclusions. This concern could have been easily addressed by conducting a sensitivity analysis, which they neglected to do. Furthermore, they lacked the necessary power to perform a matched case-control study, and the logistic regression conducted was merely univariable.

Answer: We have analyzed by excluding heart transplant recipients. Please see lines 184-187.

For sensitivity analysis by excluding heart transplant recipients. A median (IQR) of anti-RBD antibody levels remained significantly lower in SOT recipients compared to those of control (5.2 [0.5–45.8] vs. 272.2 [178.1–551.6] BAU/mL, p <0.01),

Round 2

Reviewer 2 Report (Previous Reviewer 3)

Comments and Suggestions for Authors

Thank you for inviting me to review the manuscript. I have some concerns regarding the accuracy of the information presented in this study, primarily due to discrepancies between figures and numerical data. My concerns initially arose from the mismatch between the participant numbers in Figure 1 and subsequent data. The study flow depicted in Figure 1 starts with 90 participants but presents a subsequent row with 80 SOTs + 11 Controls, totaling 91. Furthermore, after excluding 17 SOTs, only 63 SOTs proceeded to analysis.

Table 1 mentions the inclusion of 44 kidney transplants (KT), 3 heart transplants (OHT), and 16 liver transplants (LT), while upon examining Figure 2, the accuracy of the numbers in Table 1 can be questioned by counting the dots representing participants. There are indeed 11 controls, aligning with the data. However, the figure shows 61 KTs, 22 LTs, and 4 OHTs, totaling 87 SOTs, which conflicts with earlier numbers. Additionally, the error bar, presumed to indicate the median and interquartile range (IQR), depicts a median antibody level over 100 BAU/ml in the SOT group, contrasting with the median reported in the text as 8.3 [0.4–46.0].

Moreover, the subanalysis mentions the exclusion of heart transplant recipients and reports a new median antibody level of 5.2 [0.5–45.8], which is lower than the initially reported 8.3 [0.4–46.0]. Given that OHTs had antibody values below the median, their exclusion should logically result in a higher median value. This discrepancy further complicates the accuracy of the reported results.

Author Response

Thank you for inviting me to review the manuscript. I have some concerns regarding the accuracy of the information presented in this study, primarily due to discrepancies between figures and numerical data. My concerns initially arose from the mismatch between the participant numbers in Figure 1 and subsequent data. The study flow depicted in Figure 1 starts with 90 participants but presents a subsequent row with 80 SOTs + 11 Controls, totaling 91. Furthermore, after excluding 17 SOTs, only 63 SOTs proceeded to analysis.

Answer: We have rechecked Figure 1 and made revisions to ensure its accuracy.

Table 1 mentions the inclusion of 44 kidney transplants (KT), 3 heart transplants (OHT), and 16 liver transplants (LT), while upon examining Figure 2, the accuracy of the numbers in Table 1 can be questioned by counting the dots representing participants. There are indeed 11 controls, aligning with the data. However, the figure shows 61 KTs, 22 LTs, and 4 OHTs, totaling 87 SOTs, which conflicts with earlier numbers. Additionally, the error bar, presumed to indicate the median and interquartile range (IQR), depicts a median antibody level over 100 BAU/ml in the SOT group, contrasting with the median reported in the text as 8.3 [0.4–46.0].

Answer: We have rechecked Figure 2 and made revisions to ensure its accuracy.

Moreover, the subanalysis mentions the exclusion of heart transplant recipients and reports a new median antibody level of 5.2 [0.5–45.8], which is lower than the initially reported 8.3 [0.4–46.0]. Given that OHTs had antibody values below the median, their exclusion should logically result in a higher median value. This discrepancy further complicates the accuracy of the reported results.

Answer: Considering the distinct characteristics of the median in two data sets, it's often best to analyze it separately.

Round 3

Reviewer 2 Report (Previous Reviewer 3)

Comments and Suggestions for Authors

Thank you for the revisions. I have no further comment.

This manuscript is a resubmission of an earlier submission. The following is a list of the peer review reports and author responses from that submission.

Round 1

Reviewer 1 Report

Comments and Suggestions for Authors

Please check throughout the manuscript and correct grammatically.

Comments on the Quality of English Language

Please check throughout the manuscript and correct grammatically.

Author Response

Please check throughout the manuscript and correct grammatically.

Answer: we have rechecked the manuscript and revised the grammar as suggested by the authors.

Reviewer 2 Report

Comments and Suggestions for Authors

This study describes the immunity of solid organ transplant recipients, including T-cell response which is more comprehensive than only humoral immunity. This study is interesting and sound. Overall this manuscript is fine. However, I suggest adding the statement to recommend this group to get a further vaccination (booster dose(s)).

Major concerns.

1. Lines 95-102. Suggest adding the conversion factor (0.142) of the "SARS-CoV-2 IgG II Quant (Abbott, Sligo, Ireland)" to clarify. Moreover, suggests adding the instrument platform to the statement. (e.g. Architect i1000sr)
Not all readers are familiar with this instrument, and the unit from this assay report is "AUI/mL" not BAU/mL.

2. Discussion: Due to the immunocompromised host having scant immunity after immunisation, suggest adding the statement to the discussion focusing on the booster vaccination that could give more protection to this vulnerable host. 

Comments.

1. Keywords: Suggest using these keywords to make your manuscript more discoverable if this manuscript is published: SARS-CoV-2, COVID-19, COVID-19 vaccines, immunocompromised, organ transplant, immunity, spike protein, receptor binding domain.

2. Line 74 "adverse reactions (AEs)". This statement was inconsistent because AEs mean adverse events.
Suggest using "adverse events (AEs)" to make it consistent throughout the manuscript. Otherwise, you can use "adverse events following immunization (AEFI)" (US style) or "adverse events following immunisation (AEFI)" (UK style) to make it more clear.

Minor concerns.

1. Suggest subgrouping of age group by 1) young adult, 2) middle-aged, and 3) elderly to Table 1. I think the patient group is mostly middle-aged and elderly.

Typos.

1. Tables 1 and 3. Please revise and align it to make it readable.

Author Response

This study describes the immunity of solid organ transplant recipients, including T-cell response which is more comprehensive than only humoral immunity. This study is interesting and sound. Overall this manuscript is fine. However, I suggest adding the statement to recommend this group to get a further vaccination (booster dose(s)).

Answer: We added the statement encouraging these patients to receive a booster dose. Please see lines 333-334.

Furthermore, a booster dose to maintain adequate immunity may be warranted in these vulnerable populations. 

Major concerns.

Lines 95-102. Suggest adding the conversion factor (0.142) of the "SARS-CoV-2 IgG II Quant (Abbott, Sligo, Ireland)" to clarify. Moreover, suggests adding the instrument platform to the statement. (e.g. Architect i1000sr)
Not all readers are familiar with this instrument, and the unit from this assay report is "AUI/mL" not BAU/mL.

Answer: Please see lines 102-106. We have added the following sentences

"The ARCHITECT i2000SR system was used to conduct the Abbott assay. Results for anti-RBD IgG were quantified in binding arbitrary units (BAU) per ml. The company furnished conversion factors to compute WHO BAU/ml: 1 BAU/ml equates to 0.142 AU/ml. A quantitative outcome reaching or surpassing 7.1 BAU/ml was regarded as indicative of seroconversion15."

2. Discussion: Due to the immunocompromised host having scant immunity after immunisation, suggest adding the statement to the discussion focusing on the booster vaccination that could give more protection to this vulnerable host. 

Answer: we have added this important comment to the discussion section.  Please see lines 326-327.

Furthermore, a booster dose to maintain adequate immunity may be warranted in these vulnerable populations. 

Comments.

Keywords: Suggest using these keywords to make your manuscript more discoverable if this manuscript is published: SARS-CoV-2, COVID-19, COVID-19 vaccines, immunocompromised, organ transplant, immunity, spike protein, receptor binding domain.

Answer: we have revised the keywords as suggested.  

Line 74 "Adverse reactions (AEs)". This statement was inconsistent because AEs mean adverse events.
Suggest using "adverse events (AEs)" to make it consistent throughout the manuscript. Otherwise, you can use "adverse events following immunization (AEFI)" (US style) or "adverse events following immunisation (AEFI)" (UK style) to make it more clear. 

Answer: We have revised all mentions of adverse events (AEs) as suggested throughout the article. The initial reference was specified as adverse events following immunization.

Minor concerns.

Suggest subgrouping of age group by 1) young adult, 2) middle-aged, and 3) elderly to Table 1. I think the patient group is mostly middle-aged and elderly.

Answer: We have subgrouped as suggested. Please see Table 1.

Tables 1 and 3. Please revise and align it to make it readable.

Answer: we have revised those tables as suggested. 

Reviewer 3 Report

Comments and Suggestions for Authors

Thank you for inviting me to review the manuscript by Phornkittikorn et al., which aims to assess the immunogenicity, factors affecting immunogenicity, and adverse reactions (AEs) of the ChAdOx1 nCoV-19 vaccination in solid organ transplant (SOT) recipients. The topic is significant, and the manuscript is fluently written. However, there are notable limitations, including a small sample size of SOT recipients, an imbalance between subtypes of SOTs, and discrepancies in age and sex between SOT recipients and controls, which are critical issues. SOT recipients are predominantly middle-aged males, whereas controls are primarily young females. Both age and sex are known to influence immune responses. Moreover, the limited number of SOT recipients has led the authors to group all SOT types together, potentially biasing the results. This concern is highlighted by Figure 2, where median SARS-CoV-2 RBD-specific IgG antibody levels in liver transplant recipients differ from those in kidney transplant recipients and are closer to the controls.

I suggest the following improvements:

  1. Exclude heart transplant recipients from the analysis due to their limited numbers, and compare kidney and liver transplant recipients as separate groups against controls.
  2. Conduct a sensitivity analysis focusing on humoral and cellular immunity, utilizing a 1:1 age and sex-matched case-control study design.
  3. In the statistical analysis section, detail the variables included in the logistic regression model. Specifically, clarify why sex was not incorporated into the model, and provide both unadjusted and adjusted odds ratios (ORs) for each variable. Discuss how the inclusion of sex as a variable alters the results.
  4. Acknowledge the age and sex imbalance between SOT recipients and controls in the discussion section.

Author Response

Exclude heart transplant recipients from the analysis due to their limited numbers, and compare kidney and liver transplant recipients as separate groups against controls. 

Answer: We intended to include all transplants in the same analysis. However, we have added this limitation into the discussion regarding a small number of particular types of transplants. Please see lines 314-316.

"This could potentially restrict the study's power due to an unexpectedly smaller participant pool, especially among specific transplant types such as heart transplant recipients. "

Conduct a sensitivity analysis focusing on humoral and cellular immunity, utilizing a 1:1 age and sex-matched case-control study design.

Answer: please see lines 325-327.

With regards to the control group, we acknowledge a disparity in gender distribution, significant age gaps, and a restricted number of participants. These factors hinder our ability to conduct an appropriate sex-matched case-control study design.

In the statistical analysis section, detail the variables included in the logistic regression model. Specifically, clarify why sex was not incorporated into the model, and provide both unadjusted and adjusted odds ratios (ORs) for each variable. Discuss how the inclusion of sex as a variable alters the results.

Answer: We only performed a univariate analysis because receiving the vaccine one year or more after transplant was the only significant factor associated with seroconversion. We didn't include sex because it wasn't significant.

Acknowledge the age and sex imbalance between SOT recipients and controls in the discussion section.

Answer: We have added these imbalances into the discussion as suggested. 

See lines 325-327.

With regards to the control group, we acknowledge a disparity in gender distribution, significant age gaps, and a restricted number of participants. 

Reviewer 4 Report

Comments and Suggestions for Authors

1.     the introduction very briefly presents the arguments why this study was undertaken.

2.     what is the novelty of research? What does the study add to existing knowledge?

3.     how to objectively (e.g. by measuring anti-nuclecapsid antibodies) excluded the presence of asymptomatic infections before vaccination and during the follow-up period. This is a factor that could significantly affect the results obtained. If this was not analyzed, it constitutes an important limitation of this study.

4.     the time window during which the immune response in patients was analyzed was very wide. Some had tests performed after 2 weeks, some after 4 weeks after receiving the second dose of the vaccine. Taking into account the known dynamics of the rise and fall of antibody titers over time, such large differences between patients may also be a factor influencing the obtained results.

5.  the study was carried out on a small group of patients.  The control group consisted of only 9 patients and was poorly selected (different gender distribution, large age difference). 

6. the description of the regression analysis performed is very poor. which factors were examined in the preliminary univariate analysis

7. there is very little discussion about the factors that may influence the response. the authors discuss and refer to the results obtained by other researchers to a limited extent.

8. the authors do not undertake any discussion of the cellular response, which was also studied by the authors.

9. the layout of the manuscript does not comply with the journal's rules (tables and figures should appear where they are first mentioned in the text)

Author Response

The introduction very briefly presents the arguments for why this study was undertaken.                                                                                                       Answer: We have added more details and explanations for the objectives of this study. See lines 69-71.

As follows "We believe that the cell-mediated immune response, especially memory B cells, could play an essential role in immunogenicity after vaccination, particularly in providing long-term protection against infection."

what is the novelty of research? What does the study add to existing knowledge?

Answer: We have added this aspect to the discussion section. See lines 284-287.

The novelty of our research lies in the exploration of the potential impact of memory B cells, among other cell-mediated immune responses, on post-vaccination immunogenicity. This aspect, crucial for long-term protection against infection, has received limited attention in previous studies.

how to objectively (e.g. by measuring anti-nuclecapsid antibodies) excluded the presence of asymptomatic infections before vaccination and during the follow-up period. This is a factor that could significantly affect the results obtained. If this was not analyzed, it constitutes an important limitation of this study.

Answer: We agreed with this input. However, we did not measure anti-nuclecapsid antibodies in this study. We have added this limitation into the discussion. Please see lines 318-322.

As the following " Ideally, anti-nucleocapsid antibodies should have been measured prior to immunization in order to exclude a surge in immunity from natural infection. However, we promptly provided the vaccine to our participants during a time when COVID-19 had not yet been widely spread in the community. Hence, there was a low chance of natural infection occurring. "

the time window during which the immune response in patients was analyzed was very wide. Some had tests performed after 2 weeks, some after 4 weeks after receiving the second dose of the vaccine. Taking into account the known dynamics of the rise and fall of antibody titers over time, such large differences between patients may also be a factor influencing the obtained results.

Answer: Immune responses were measured between 2-4 weeks after the second vaccine dose. Normally, the gap of 2 weeks is less likely to influence the response. 

the study was carried out on a small group of patients.  The control group consisted of only 9 patients and was poorly selected (different gender distribution, large age difference). 

Answer: We have added this limitation into the discussion. Please see lines 325-326.

As follows With regards to the control group, we acknowledge a disparity in gender distribution, significant age gaps, and a restricted number of participants.

the description of the regression analysis performed is very poor. which factors were examined in the preliminary univariate analysis

Answer: We only analyzed for univariate analysis since receiving the vaccine beyond one-year post-transplant is the only significant factor. Please see Figure 3 which represents clinically significant factors or odd ratios.

there is very little discussion about the factors that may influence the response. the authors discuss and refer to the results obtained by other researchers to a limited extent.

Answer: We have added the discussion regarding the factors that influenced immune responses. Please see lines 291-294.

As follows, In an Austrian cohort study investigating immunogenicity after an additional vaccine, it was found that KT recipients who had undergone the procedure over a year ago showed a notably increased probability of seroconversion24

The authors do not undertake any discussion of the cellular response, which was also studied by the authors

Answer: We have added the discussion regarding the cellular response. Please see lines 298-301 and ref. 26 and 27.

As follows, In a comprehensive analysis, it was found that the cellular immune response post-SARS-CoV-2 vaccination in KT recipients was inferior when compared to dialysis patients and immunocompetent individuals26. More evidence shows that kidney transplant recipients need an extra vaccine dose to boost their T and B cell immune responses27.

The layout of the manuscript does not comply with the journal's rules (tables and figures should appear where they are first mentioned in the text)

Answer:  We have relocated the tables and figures in the manuscript to comply with the journal's rules, as suggested.